Manuscript prepared for Atmos. Chem. Phys.
with version 2014/09/16 7.15 Copernicus papers of the LATEX class copernicus.cls.
Date: 20 June 2018

# Status and future of Numerical Atmospheric Aerosol Prediction with a focus on data requirements

Angela Benedetti[1], Jeffrey S. Reid[2], Peter Knippertz[15], John H. Marsham[6,17],
Francesca Di Giuseppe[1], Samuel Rémy[5], Sara Basart[4], Olivier Boucher[5], Ian
M. Brooks[6], Laurent Menut[18], Lucia Mona[19], Paolo Laj[16,25],
Gelsomina Pappalardo[19], Alfred Wiedensohler[23], Alexander Baklanov[3],
Malcolm Brooks[7], Peter R. Colarco[8], Emilio Cuevas[9], Arlindo da Silva[8],
Jeronimo Escribano[5], Johannes Flemming[1], Nicolas Huneeus[10,11], Oriol Jorba[4],
Stelios Kazadzis[12,13], Stefan Kinne[14], Thomas Popp[20], Patricia K. Quinn[24],
Thomas T. Sekiyama[21], Taichu Tanaka[21], and Enric Terradellas[22]

[1]European Centre for Medium-Range Weather Forecasts, Reading, UK
[2]Naval Research Laboratory, Monterey, CA, USA
[3]World Meteorological Organisation, Switzerland
[4]Barcelona Supercomputing Center, BSC, Barcelona, Spain
[5]Institut Pierre-Simon Laplace, CNRS / Sorbonne Université, Paris, France
[6]University of Leeds, Leeds, UK
[7]UK Met Office, Exeter, UK
[8]NASA Goddard Space Flight Center, Greenbelt, Maryland, USA
[9]Izaña Atmospheric Research Centre, AEMET, Santa Cruz de Tenerife, Spain
[10]Geophysics Department, University of Chile, Santiago, Chile
[11]Center for Climate and Resilience Research (CR)2, Santiago, Chile
[12]Physikalisch-Meteorologisches Observatorium Davos, World Radiation Center, Switzerland,
Davos, Switzerland
[13]National Observatory of Athens, Greece
[14]Max-Planck-Institut für Meteorologie, Hamburg, Germany
[15]Karlsruhe Institute of Technology, Karlsruhe, Germany
[16]univ. Grenoble-alpes, IGE, CNRS, IRD, Grenoble INP, Grenoble, France
[17]National Centre for Atmospheric Science, UK
[18]Laboratoire de Météorologie Dynamique, Ecole Polytechnique, IPSL Research University, Ecole
Normale Supérieure, Université Paris-Saclay, Sorbonne Universités, UPMC Univ Paris 06, CNRS,
Palaiseau, France
[19]Consiglio Nazionale delle Ricerche, Istituto di Metodologie per l'Analisi Ambientale
(CNR-IMAA), C. da S. Loja, Tito Scalo (PZ), Italy
[20]German Aerospace Center (DLR), German Remote Sensing Data Center Atmosphere,
Oberpfaffenhofen, Germany
[21]Japan Meteorological Agency/Meteorological Research Institute, Tsukuba, Japan
[22]Spanish Meteorological Agency, AEMET, Barcelona, Spain
[23]Leibniz Institute for Tropospheric Research, Leipzig, Germany
[24]National Oceanic and Atmospheric Administration, Pacific Marine Environmental Laboratory,
Seattle, USA
[25]Department of Physics, University of Helsinki, Helsinki, Finland

*Correspondence to:* Angela Benedetti (A.Benedetti@ecmwf.int)

**Abstract.** Numerical prediction of aerosol particle properties has become an important activity at many research and operational weather centres. This development is due to growing interest from a diverse set of stakeholders, such as air quality regulatory bodies, aviation and military authorities, solar energy plant managers, climate services providers, and health professionals. Owing to the complexity of atmospheric aerosol processes and their sensitivity to the underlying meteorological conditions, the prediction of aerosol particle concentrations and properties in Numerical Weather Prediction (NWP) framework faces a number of challenges. The modeling of numerous aerosol related parameters increases computational expense. Errors in aerosol prediction concern all processes involved in the aerosol life cycle including a) errors on the source terms (for both anthropogenic and natural emissions); b) errors directly dependent on the meteorology (e.g., mixing, transport, scavenging by precipitation); c) errors related to aerosol chemistry (e.g., nucleation, gas-aerosol partitioning, chemical transformation and growth, hygroscopicity). Finally, there are fundamental uncertainties and significant processing overhead in the diverse observations used for verification and assimilation within these systems. Indeed, a significant component of aerosol forecast development consists in streamlining aerosol related observations and reducing the most important errors through model development and data assimilation. Aerosol particle observations from satellite and ground-based platforms have been crucial to guide model development of the recent years, and have been made more readily available for model evaluation and assimilation. However, for the sustainability of the aerosol particle prediction activities around the globe, it is crucial that quality aerosol observations continue to be made available from different platforms (space, near-surface, and aircraft) and freely shared. This paper reviews current requirements for aerosol observations in the context of the operational activities carried out at various global and regional centres. While some of the requirements are equally applicable to aerosol-climate, the focus here is on global operational prediction of aerosol properties such as mass concentrations and optical parameters. It is also recognized that the term "requirements" is loosely used here given the diversity in global aerosol observing systems and that utilized data are typically not from operational sources. Most operational models are based on bulk schemes that do not predict the size distribution of the aerosol particles. Others are based on a mix of "bin" and bulk schemes with limited capability to simulate the size information. However the next generation of aerosol operational models will output both mass and number density concentration to provide a more complete description of the aerosol population. A brief overview of the state-of-the-art is provided with an introduction on the importance of aerosol prediction activities. The criteria on which the requirements for aerosol observations are based are also outlined. Assimilation and evaluation aspects are discussed from the perspective of the user requirements.

# 1 Introduction

Over the last two decades, the concept of global observing systems and the importance of defining user requirements for the purpose of monitoring and forecasting elements of the Earth System have gained momentum. This also applies to atmospheric composition in general and aerosol in particular with the studies of **?** for atmospheric composition monitoring, **?** for operational aerosol forecasting, **?** for operational verification of aerosol properties, and **?** on the use of Earth Observing System data for aerosol operational systems. Indeed, at the time of writing this document, there are at least nine operational centers producing and distributing real-time global aerosol forecasting products, including: ECMWF Copernicus Atmosphere Monitoring Service (CAMS), Finnish Meteorological Institute (FMI), Fleet Numerical Meteorology and Oceanography Center (FNMOC), Japan Meteorological Agency (JMA), NOAA National Center for Environmental Prediction (NCEP), and UK Met Office. In addition, there are numerous quasi-operational centers generating near real time data streams and forecasts, including, Barcelona Supercomputing Center (BSC), Météo-France, and NASA's Global Modeling and Assimilation Office (GMAO). Each of these centers has its own internal requirements for data to support data assimilation, evaluation, development and ultimately user specific product delivery of their aerosol forecasting programs. Commissioned by the World Meteorological Organization (WMO), this document outlines the requirements of the aerosol prediction system developers (the data "users" in this context). It has been compiled through consultation with experts in aerosol modeling, assimilation and evaluation both from the operational centers and the aerosol research community. However, it is recognized from the onset that compositional forecasting evolved is in its infancy relative to its well matured Numerical Weather Prediction (NWP) parentage, with a high dependence on non-operational data sources and diversity in modeled parameters and architecture. Even functional definitions differ between developers. At the same time, the compositional community is aware of mainstream NWP's own requirements challenges for observations, architecture, distribution, formats, quality assurance etc., all with far fewer degrees of freedom than the atmospheric composition community faces. Therefore we see this document as the beginning of an evolutionary process towards more specific technical requirements in the future.

## 1.1 Context and needs of the numerical atmospheric composition prediction community

Numerical atmospheric aerosol prediction (NAAP) is still an activity in its infancy, borne largely from the global climate and air quality communities. It is a sub-component of the larger and far more mature field of NWP, and as such, it is reasonable to expect that NAAP will follow the overall architecture and best practices set up by the NWP community. This includes in particular best practices in using and setting requirements for observational data. Just as there are requirements for radiosonde releases and weather station data transmission, one would expect similar considerations for parameters such as $PM_{10}$ (total mass of particles with aerodynamic diameter less than 10 $\mu$m),

PM$_{2.5}$ (total mass of particles with aerodynamic diameter less than 2.5 $\mu$m) and other key parameters such as Aerosol Optical Depth (AOD), extinction coefficient, mass concentrations of individual chemical components, and light scattering and absorption coefficients. To a large degree this type of data is already being collected in many countries around the world and inter-calibration procedures are in place in existing surface networks. This said, it is acknowledged that even within say the typical WMO meteorological feeds there are differences in reporting practices between countries, long standing biases between instrumentation deployed, and challenges to modernization (e.g., in commercial radiosonde products, **?**). There are, however, a number of additional unique challenges facing the NAAP community that should be addressed and integrated in the development of relevant global aerosol data streams. There is a long history of reporting and sharing meteorological data because it is understood to be of mutual benefit to all parties in the exchange and, weather being considered an "act of nature" there is less political motive behind data policies. Atmospheric composition data, however, is often related to air quality through anthropogenic emissions of pollutants and thus has local regulatory or even international treaty ramifications. There can subsequently be some local hesitance to report unfavourable data, or at the least to provide additional funding to ease its distribution. One exception is dust storms, and indeed reporting of dust observation and prediction is more mature than any other aerosol species, even though there are only a few ground stations in key source areas. Even so, the enhancement of dust production due to water policy decisions can be divisive. Compositional data collection can be far more expensive in equipment and analytical services, and, often, difficult to calibrate. While NWP has suffered at times with diversity in, for example, commercial radiosonde providers and instrument efficacy (e.g., relative humidity), aerosol measurement has considerably more degrees of freedom in its measurement technology, overall maintenance, and reporting. Indeed, significant diversity exists in composition measurements including chemistry and size related parameters, in particular in regard to carbonaceous species and the coarse mode, respectively (**????**). While institutions such as the World Meteorological Organization (WMO), the United States Environmental Protection Agency (EPA) or the European Environment Agency (EEA) set benchmark levels for air quality monitoring, they are by no means universally applied. The research community is nevertheless making a huge effort to intercompare and standardize their measurements. However, it is another step for standardization to be universally applied. Furthermore, the ability to report with a given timeliness, critical for NAAP and NWP consumption alike is related to measurement technology. A host of potential variables can be generated relating to mass, composition, optical properties, or microphysics. Deployed instruments and their locations are also constantly evolving. The authors of this paper are keenly aware of the difficulties associated with aerosol measurements and the efforts made to improve these. The "requirements" or recommendations made herein should not be interpreted as criticisms of the existing observing system but rather an acknowledgement of the current state of the field and as a mean to move forward. They are not meant to introduce more rigidity but rather should be interpreted for awareness and practicality.

Given the early state of the field and diversity in development approaches and customer requirements at aerosol prediction centres, the community requires flexibility as it finds its way. Regardless of data type, whether in situ or from remote sensing, there are three guiding principles that should be considered.

1. Data should be easily accessible, publicly available, reasonably well documented, and for baseline quantities, encoded into a similar format. Currently data distribution is diffuse and potential users have difficulty maintaining and evaluating global scale data outside of the largest and most consistent networks (for example the NASA Aerosol Robotic Network-AERONET sun photometer dataset, **?**). While long term sites are preferred, the operational reality has been for a reduction in support for key supersites, such as Atmospheric Radiation Measurement (ARM) or Global Atmospheric Watch (GAW). Thus, future data distribution models could mimic meteorological data, where observations are broadcast and consolidated for use (e.g., 6 or 12 hourly $PM_{2.5}$ or $PM_{10}$ data). However, care must be taken to avoid ongoing legacy issues in the current broadcast system.

2. Timeliness requirements also vary by center. Based on the consensus of centers, 3 hour latency is preferred, and 6 hours is adequate, especially for satellite products. There is nevertheless value in 12 hour or even multi day delivery for evaluation and model refinement purposes, including surface particulate matter monitoring. Timeliness should be a goal, but not necessarily a requirement. This is especially true for compositional data requiring laboratory work for analysis.

3. Realistic error bars or error models must be provided. The operational community can easily cope with uncertain data, provided that uncertainty is known on a data point by data point basis. Indeed, error tolerances are strongly customer and application related.

Mindful of these considerations, specific issues and definitions of user requirements are addressed in the following subsections. Note that in this paper no mention is made of the volcanic ash aerosol system. While the prediction of this type of aerosol is essential for numerous applications, we believe that there is a need for a separate study dealing with specific requirements for volcanic ash aerosols. Several communities are dealing with this topic, for example the Global Atmospheric Watch (GAW) Scientific Advisory Group (SAG) on Volcanic Ash, the GAW SAG on Applications, the aerosol lidar networks and their confederation (e.g., MicroPulse Lidar NETwork-MPLNET, European Aerosol Research Lidar Network-EARLINET, GAW Aerosol Lidar Observation Network-GALION), and others. The AEROSOL Bulletin 3 available from WMO provides an overview of current efforts on this topic (available from WMO, https://library.wmo.int).

## 1.2  The nature of user requirements

The notion of user requirements implies that the specific technology or science application has an underlying group or community that has an interest in using the data, be it data from an observational platform or simulations from a model. Communities use the data for their applications, and this (implicitly or explicitly) sets the requirements. One of the principles behind the development of user requirements implies that data requirements should be put forward by the relevant commu-

nities independently of the current technologies and systems available, with the overarching goals of supporting the applications of the community in question, for example weather prediction, ocean modeling, climate investigation etc. Specifically for observation requirements, no consideration is given to what type of instruments, observing platforms or data processing systems are necessary or even possible to meet them. Even though in practice, it is not possible to make user requirements

completely technology-free and current availability of technology influences their formulation, it is a useful exercise to understand data gaps and also to establish if new observing systems can meet all or part of the user requirements. This process of formulating user requirements establishes also an important direct link between model developers and data providers. Many data products that are provided by environmental agencies or individual scientists, end up not being in the model develop-

ment/ assimilation/ assessment loop as they do not correspond with what is needed by the modelers (e.g., in terms of accessibility, timeliness, quality, or uncertainty). Vice versa, often model developers have unrealistic expectations, do not specify their priorities and end up using only a sub-set of available observations. Dialogue between these two communities is what ultimately fosters progress on both sides. The requirements for observations are usually given in terms of the following criteria:

(i) resolution (horizontal and vertical and sometimes temporal), (ii) sampling (horizontal and vertical), (iii) frequency (how often a measurement is taken in time), (iv) timeliness (i.e., availability), (v) repetition cycle (how often the same area of the globe is observed), and most importantly (vi) uncertainty either related to the actual instrument accuracy and/or to the algorithm used to perform the retrieval in case of derived observations (for example aerosol optical depth or total column ozone).

Additionally, the user must specify what physical or chemical variables should be measured.

Resolution and sampling differ in that resolution relates to the area and time period a measurement is representative of, whereas sampling indicates the distance between two successive measurements both in space and time. Frequency is related to the temporal sampling of an instrument whereas repetition gives a measure of how often the same location is observed. For example, an instrument

on a polar orbiting satellite may have very high frequency but low repetition.

Uncertainty can be divided into accuracy, which relates to the bias of the measurement, and precision, which relates to the random error. For example in the presence of biased observations, averaging more observations does not generally improve the accuracy, but may improve the precision. For each application, it is generally accepted that improved observations in terms of resolution, sam-

pling, frequency and accuracy, etc. against some baseline are generally more useful than coarser, less

frequent and less accurate counterpart observations. The latter, however, could still be useful. Some of the criteria may come into play depending on the particular area of application. For example, timeliness is a criterion which is not included in the requirements for climate research whereas due to the constraints on the timely delivery of the forecasts, it is a crucial parameter for operational prediction and assimilation. The usefulness of an observation is dependent on the specific application and its availability. This is specified in the requirements by adding three values per criterion: the "goal", the "threshold" and the "breakthrough". The goal is the value above which further improvement of the observation would not bring any significant improvement to the application. Goals may evolve depending on the progress of the application and the capacity to make better use of the observations. The threshold is the value below which the observation has no value for the given application. An example of threshold requirement for assimilation is, for example, the timeliness of the data: observations that are delivered beyond a certain time (normally three to six hours for near-real time NWP applications) cannot be used in the analysis. The breakthrough is a value in between the goal and the threshold that, if achieved, would result in a significant improvement for the application under consideration. Of these three parameters the most elusive is the breakthrough because its value may change more drastically than the other two with system developments.

While the usefulness class of requirement is conceptually straightforward it is less so functionally and consequently can have an arbitrary nature in a rapidly developing field such as NAAP. Thus, while this document will provide examples of usefulness, there is a hesitation to be overly specific at this time. In particular breakthrough and goal values for different variables are not independent: accurate measurements of one variable may lower the usefulness of another less accurately-measured variable because the variables are related in the model. For instance AOD measurements become less valuable if measurements for surface monitoring if the full profile of the extinction coefficient becomes available with the required sampling and accuracy.

## 1.3 Rolling Review of Requirements and Task Team on Observational Requirements and Satellite Measurements

The WMO has developed a framework for different thematic areas such as Global Numerical Weather Prediction, High-resolution Numerical Weather Prediction, Nowcasting and Very Short Range Forecasting, Ocean applications, and Atmospheric Chemistry, among others, to be reviewed periodically in terms of design and the implementation of various observing systems, using as guidance the user requirements set-out by the relevant community (**?**). This process is called the rolling review of requirements (RRR) and it involves several steps. For each application area, these steps are as follows: (i) a review of "technology-free" user requirements (i.e., not taking into account the available technology) for observations in one of the thematic areas; (ii) a review of current and future observing capabilities (space-based and surface-based); (iii) a critical review of whether the capabilities meet the requirements; and finally (iv) a statement of guidance based on the outcomes of the criti-

cal review. This statement of guidance is often called gap analysis as it shows whether the current observing system is suitable for the given application and what is needed in the future observing system in order for it to meet the requirements set out by the user community. To facilitate this pro-

cess, the WMO maintains an online database on user requirements and observing system capabilities called Observing Systems Capability Analysis and Review tool (OSCAR). Details on the RRR are provided in **?**, and references therein.

Recently, the WMO GAW set up an ad-hoc Task Team on Observational Requirements and Satellite Measurements as regards Atmospheric Composition and Related Physical Parameters (TT-

ObsReq, http://www.wmo.int/pages/prog/arep/gaw/TaskTeamObsReq.html) to review the user re-quirements specifically for atmospheric composition. Application areas related to atmospheric com-position include: (i) Forecasting Atmospheric Composition which covers applications from global to regional scales ($\approx$10 km and coarser) with stringent timeliness requirements (NRT) to support op-erations such as sand and dust storm and chemical weather forecasts, (ii) Monitoring Atmospheric

Composition which covers applications related to evaluating and analysing changes (temporally and spatially) in atmospheric composition regionally and globally to support treaty monitoring, clima-tologies and re-analyses, assessing trends in composition and emissions/fluxes, and to better un-derstand processes, using data of controlled quality (and with less stringent time requirements than needed for NRT). (iii) Providing Atmospheric Composition information to support services in urban

and populated areas, which covers applications that target limited areas (with horizontal resolu-tion of a few km or smaller) and stringent timeliness requirements to support services related to weather/climate/pollution, such as air quality forecasting.

The WMO GAW TT-ObsReq analyzed the role of atmospheric composition observations also in support of the other WMO application areas (http://www.wmo.int/pages/prog/www/OSY/GOS-

RRR.html). After the Second Workshop of the TT-ObsReq (12-13 August 2014, Zurich), the com-mittee identified key parameters needed for Forecasting Atmospheric Composition. For aerosols these parameters were: aerosol mass, size distribution (or at least mass in three fraction sizes: up to 1, 2.5 and 10 micron as it is common practice in air quality, speciation and chemical composition, AOD at multiple wavelengths, absorption AOD (AAOD), ratio of vertically integrated mass to AOD,

vertical distribution of aerosol extinction). Some of the parameters outlined for Monitoring Atmo-spheric Composition may also be relevant to the operational prediction of aerosol particle properties, which is one of the application areas (Forecasting Atmospheric Composition) and is the focus of this study. Because recommendations from the committee are technology-free, they differ slightly from those identified by the Scientific Advisory Group on Aerosol (GAW report 227), which limits their

recommendations to variables that can be directly measured.

Requirements are outlined based on what is needed for the fundamental components of an aerosol prediction system which are: (i) modeling processes (aerosol particles emission, secondary produc-tion and removal), (ii) data assimilation (when present), and (iii) model evaluation. Section 2 briefly

presents current operational and pre-operational aerosol systems both at the global and the regional
scales. Section 3 describes the data needs and the requirements for emission and removal processes,
section **??** outlines those for the assimilation component, and finally section **??** describes those re-
lated to model evaluation. Section **??** summarizes those data needs and includes some final thoughts.

## 2   Aerosol Prediction Models

Several centres with operational or quasi-operational capabilities are currently running aerosol pre-
diction systems. These are BSC, ECMWF, FMI, FNMOC/NRL, GMAO, JMA, Météo-France, NCEP,
UK Met Office on the global level. There are also numerous regional models run by the above cen-
tres as well as for example the Chinese Meteorological Agency (CMA), the Korean Meteorological
Agency (KMA), the Institut national de l'environnement industriel et des risques (INERIS), the
Deutscher Wetterdienst (DWD), just to mention a few. These systems are used for various applica-
tions, including, but not exclusively, global air quality forecasts (dust and biomass burning), opera-
tion impacts, boundary conditions for regional systems and flight campaign planning (**?**). Each relies
on different dynamical cores, advection solvers, and aerosol microphysics schemes that necessarily
generate a large degree of diversity among the various models (see for example **?**). The range of
horizontal and vertical resolutions across the models is also very diverse, as is inline versus offline
architecture. In general, increasing resolution does not necessarily mean better model skills as it may
request new tuning of parameters of subscale processes (e.g. orographic gravity wave drag), as well
as larger ensemble runs due to high variability. While all centers are pursuing data assimilation, four
have multiple species data assimilation capabilities (namely ECMWF, FNMOC/NRL, GMAO, and
JMA), while UKMO has a dust only system with data assimilation. Methods in development vary
from 2D-Var, 4D-Var, EnKF and hybrid schemes.

In recent years, aerosol forecasting centres have been turning to ensemble prediction to describe
the future state of the aerosol fields from a probabilistic point of view. Multi-model consensus prod-
ucts have been developed to alleviate the shortcomings of individual aerosol forecast models while
offering an insight on the uncertainties and sensitivities associated with a single-model forecast. Ex-
amples include the International Cooperative for Aerosol Prediction (ICAP) Multi Model Ensemble
(ICAP-MME, **?** (http://www.nrlmry.navy.mil/aerosol/) for global aerosol forecasts and the WMO
Sand and Dust Storm Warning Advisory and Assessment System (SDS-WAS) North African and
Middle East Regional Node for regional dust forecasting (http://sds-was.aemet.es/; **?**). Both initia-
tives have demonstrated that simply collecting different forecasts in a single database and generat-
ing web pages with common plotting conventions is an effective tool for developers to assess and
improve their forecasting systems. Use of ensemble forecast techniques is especially relevant for
situations associated with unstable weather patterns, or in extreme conditions. Ensemble approaches
are also known to have more skills at longer ranges (> 3 days) where the probabilistic approach pro-

vides more reliable information than a single model run due to the model error increasing over time. Moreover an exhaustive comparison of different models with each other and against multi-model products as well as observations can reveal weaknesses of individual models and provide an assessment of model uncertainties in simulating the aerosol cycle. Multi-model ensembles also represent a paradigm shift in which offering the best product to the users as a collective scientific community becomes more important than competing for achieving the best forecast as individual centres. This new paradigm fosters collaboration and interaction, and ultimately results in improvements in the individual models and in better final products.

A detailed description of the individual models is beyond the scope of this paper. For a review of the current systems that provide aerosol forecasts, some with focus on dust, see for example **?** and **?**. Ensemble systems are presented in **?** and **?**. An overview of regional aerosol forecasting systems can be found in **?????**. In the rest of the paper, we will mainly focus on requirements for global models, acknowledging that regional (i.e., limited-area) models may have different sets of requirements, including additional boundary conditions. Regional ground-based networks can for example address some of those needs while not providing sufficient coverage for global models (e.g., AERONET DRAGON networks, **?**). Global observations can be of use also for regional applications but the requirement on the resolution, for example, may differ from that of a global model. In general most of the requirements below will apply to both global and regional models. Moreover, although some of the data requirements presented here are shared with aerosol models for climate applications, here we focus on numerical aerosol prediction at the short and medium-range (up to 10 days). In this context we are essentially dealing with an initial and boundary condition problem for which the requirements for assimilation have high importance. For sub-seasonal to seasonal aerosol prediction, which is not dealt with here specifically, requirements on ocean state and variability are also important as well requirements for the development of prognostic emission models. In the wider context of aerosol projections for climate prediction, the emphasis is much more on emission scenarios and the requirements will consequently be different.

## 3 Modeling of aerosol particle emission and removal processes

### 3.1 General concepts

Modeling of aerosol particle sources and sinks are of uppermost importance, because these processes largely control the spatio-temporal distributions of aerosol particle concentrations and size distributions. In addition, in polluted environments, uncertainties are dominated by emissions whereas in remote regions transport and aerosol processes control the uncertainty . For a given source strength, sinks also control the atmospheric residence times of aerosol particles, which is in turn a key indicator of long-range transport of aerosol species. A good representation of aerosol particle sources and sinks is particularly important to determine the overall analysis and forecast of particle mass,

surface area, and number concentrations in regions with few observations for data assimilation. A
discrepancy in aerosol sources and/or sink processes can cause a systematic drift in aerosol particle
concentrations and AOD over the forecast range in a forecasting system with data assimilation. This
is because often the data assimilation corrects for the bias in sources and/or sinks. This correction is
often not retained in the subsequent forecast integration due to the fact that the model does not repre-
sent the emission/removal processes adequately. For this reason, it is useful to set user requirements
also for source and sink observations of aerosol particles. Efforts to formulate aerosol data assimi-
lation with emission fluxes as well as or instead of mixing ratios as a control variable might have a
role to play in correcting these forecast drifts, although such observations would remain important
constraints in such a framework.

It is appropriate to differentiate sources of aerosols and aerosol precursors that are directly emit-
ted by human activities from those (natural or anthropogenic) emissions that depend on natural pro-
cesses. User requirements for directly-emitted anthropogenic emissions can be articulated around
the following criteria: accuracy, spatial resolution, temporal resolution, speciation, aerosol size dis-
tribution and hygroscopicity. User requirement for emissions that depend on meteorological pro-
cesses also include requirements on key meteorological and environmental quantities that control
such emissions, for example winds and surface conditions or any other parameters that may lead to
aerosol formation.

### 3.2   User requirements for desert mineral dust emissions

For a reliable prediction of mineral dust aerosol, sufficiently accurate knowledge of both the emitting
soil and the deflating winds is needed. Both aspects suffer from insufficient observational constraints,
creating a large challenge for quantitative emission predictions. Important source regions globally
include the Sahara/Sahel, Southwest Asia/Middle East, Taklimakan/Gobi deserts of China, Aus-
tralia and the Southwest United States/adjacent Mexico (**?**). However, larger source regions show
substantial fine structure and throughout the world there are also many individual sources such as in
Patagonia, the Arctic plains, and countless dry or drying lake beds. Estimating dust emission sources
can also be performed from satellite data (for examples see **??????**).

Dust models typically employ maps of dust source functions (e.g. (**??**)), because soil properties in
arid and hyper-arid regions from global inventories are insufficient to provide consistent soil texture
information. This includes aspects such as soil particle size distribution and binding energies but also
the existence of roughness elements and soil moisture content that impact on mobilization thresh-
olds. See **?** for a comprehensive review. This severely limits the level of complexity that can be put
into models representing the physical processes of dust emission (e.g. **???**). In order to get a better
understanding of the involved uncertainties, an update to the objective comparison of different dust
source inventories by **?** would be desirable and could be extended to take into account uncertainties
in the dust emission parameterization itself.

In addition to that, dust emission is further complicated by suppressing influences of soil moisture
       (**?**) and vegetation cover, including brown vegetation from a previous rainy period (**?**), which can
       vary on relatively small time and spatial scales. This is particularly acute in the semi-arid Sahel with
       its seasonal vegetation, also creating large variations in surface roughness (**?**). There is currently a
       debate to what extent the mineralogy of emitted dust particles should be taken into account, as this
would alter both its interactions with radiation (**?**) and cloud microphysics (**?**). While certainly this
       is an interesting field of research, the former aspect is probably more relevant on longer timescales,
       and the latter is not even considered in most current dust prediction models.

       Surface wind speeds, particularly peak gusts, are also poorly represented in many meteorological
       models (**?**) and this induces errors both in dust emissions and subsequent transport (**?**). Indeed,
given the strong nonlinearity in dust production to wind, the gusts may dominate the nature of dust
       production (e.g., **?**). This may be particularly true for northern Africa but many aspects apply to other
       source regions around the world, too. For example, many models create too much vertical mixing
       in the stable nighttime planetary boundary layer over arid areas, leading to an underestimation of
       nocturnal low-level jets and a too flat diurnal cycle in surface winds (**?**; **?**; **?**). This is partly related
to an underestimation of turbulent dust emission during the day (**?**). Another substantial problem is
       the lack of dust generation related to cold pools (haboobs) associated with moist convection over the
       Sahel and Sahara (and many other desert areas in Asia, Australia and America), a process largely
       absent in models with parameterized convection (**?**; **?**; **??**). This leads to even reanalyses missing the
       summertime maximum in dust generating winds in the central Sahara (**??**).

It is challenging to improve model representation of dust generation due to an enormous lack of
       observations from key source regions. The logistically difficult and politically unstable Saharan and
       Middle East region has large areas void of any ground stations. What is required to better understand
       and specify the meteorology of dust production, is a much denser network of stations that observe
       standard meteorological parameters such as wind, temperature, humidity and pressure, ideally lo-
cated in some of the main source regions. Given the large diurnal cycle and the short lifetime of
       some dust-raising mechanisms, particularly moist convection, an hourly or better time resolution
       would be desirable (**??**). A first step in creating such a network was undertaken during the recent
       Fennec project, which deployed stations in 2011 (**?**), but the deployed stations could not be main-
       tained beyond 2013 (**?**), so do not provide continuous monitoring or a long climatology, but have
demonstrated that: (i) reporting the sub 3-minute variance in winds is generally unimportant, but
       resolving the diurnal cycle is critical, (ii) there are substantial biases even in analyzed winds, which
       miss the summer-time wind maximum in the central Sahara, and (iii) that it is important to eval-
       uate dust uplift together with model winds, and that observational records of this relationship are
       invaluable (**?**).

The lack of observations in combination with the difficult-to-represent meteorology also leads to
       substantial deviations between different analysis products, even on continental scales (**?**), creating

substantial differences in dust emission (e.g. **?**). Yet, the fine scale nature of dust emissions prevents large scale observations from providing constraint into what a "correct" dust source function is; rather available observations provide only a gross tuning parameter (**?**). Particularly the depth of the Saharan heat low, which is crucial for the large-scale circulation over northern Africa and thus a dominating factor for dust generation, can vary substantially between different analyses or model simulations with different resolution (**?**). A much denser network of high-quality pressure and wind observations is needed to better constrain models in this regard. Pressure measurements have the advantage of being less affected by local conditions (e.g. topographic circulations, inhomogeneities in roughness) than wind measurements, and have – through data assimilation – a far greater impact on the analyzed heat low, which in turn controls the model winds. Although, direct wind measurements over under-observed source regions would also be highly desirable.

In addition, our knowledge of the amount and the size distribution of the emitted mineral dust particles is limited. Significant diversity exists between measurement methods for airborne dust (**?**), with aerodynamic and inversion methods being generally in agreement (**?**), and with optical particle counters showing larger sizes. This leaves mass as one of strongest constraints on the system. Investment is required in instrumentation that can accurately characterize coarse and giant aerosol particles. A network of ground stations is subsequently required that in addition to standard meteorology measures mineral dust emission, ideally including mass or number size distributions of emitted particles. Ideally such stations should be complemented with information about the state of the soil (texture, soil moisture, vegetation, mineralogy). Some such efforts were made during recent field campaigns such as Fennec (**?**), the Bodélé Dust Experiment (BoDEx) (**?**) and the Japanese Australian Dust Experiment (JADE) (**?**) to just name a few examples. Longer-term monitoring stations, however, are very rare, with the African Monsoon Multidisciplinary Analysis (AMMA) Sahelian Dust Transect (SDT) being a notable exception (**??**). Worth mentioning are also the CV-DUST project (**?**) and the Cape Verde Atmospheric Observatory (CVAO) with its long term dust record (**?**). An extension of such activities to more remote source areas would be highly desirable, especially one that accounted for large particles.

Given the relative lack of in situ data a continued reliance on remote sensing is anticipated in coming years, but a number of challenges remain. First, obscuration of dust by cloud (**?**) is likely a problem that cannot be solved. Second, much summertime dust is emitted at night (**?**) but most current products are day-time only, requiring better information from wavelength other than visible. Infrared products from geostationary satellites are being developed but still have biases related to atmospheric moisture and uncertainties from the dust optical properties (**??**). These would need to be further improved and provided in near-real time for data assimilation, but have been useful for source detection (**?**). Newly developed dust optical depth products such as those from infrared high-spectral sensors (e.g., Infrared Atmospheric Sounder Interferometer (IASI) (**???**)) or those produced with the GRASP algorithm (**?**) are promising but have more limited space-time coverage. In addition,

location of AERONET stations closer to source regions (as discussed in **?**) would allow evaluation
of models and satellite retrievals near source (e.g., the short-term deployment during the Fennec
field campaign **?**), and retrievals from such observations should in future account for particles with
diameters exceeding 30 $\mu$m (**?**).

Lidar technique advancements occurred in the last decade allow nowadays to have a better insight
of the desert dust distribution in the atmospheric columns. Respect to the conventional passive re-
mote sensing techniques, lidar measurements provide optical properties of the atmospheric aerosol
as a function of the altitude. This implies that aerosol layers can be identified and characterized in
terms of optical properties by lidar measurements. Different lidar techniques exists with different
levels of accuracy, but their added value for desert dust observations in measurement campaigns,
long term measurements and model evaluation is widely demonstrated (see **??** for more details).
In addition, depolarization measurement capability allows reliable identification of non-spherical
particle presence and therefore a reliable information on the contribution of the desert particle to
the aerosol backscatter and extinction coefficient as a function of the altitude. Desert dust profiles
provided by CALIPSO at global level since 2006 improved our knowledge of the desert dust distri-
bution in the atmospheric column and on the transport mechanisms and impacts world-wide (e.g. **?**)
. At ground-based level, lidar networks like EARLINET, MPLnet and ADnet are improving more
and more in terms of methodologies and observational capability, fostering also the link and synergy
with more operational communities like the ceilometer one. This said, characterization of the the
near surface environment is problematic, with attenuation being an issue for space and airborne li-
dars, and overlap corrections for lidars at the surface looking upwards. Regardless, the advancements
in lidar observations is going to improve the overall knowledge of the desert dust vertical distribu-
tion in particular close to the source regions through satellite measurements (CALIPSO, the Cloud
Aerosol Transport System (CATS), to a limited extent ESA Doppler wind lidar ALADIN on Aeolus
and to a fuller extent ATLID on EarthCARE) and low cost automatic systems like ceilometers.

Finally, the dust-focused satellite data should be complemented by improved space-born assess-
ments of soil moisture, vegetation cover (green and brown) and soil mineralogy to better characterize
varying conditions in source regions (**?**). For soil mineralogy, airborne and space-borne spectroscopic
mapping (as DLR-EnMap and upcoming NASA-EMIT missions) provides a new resource to deter-
mine the relative abundance of the key dust source minerals with sufficient detail and coverage, but
this resource has been virtually unexplored in the context of dust modeling.

## 3.3 User requirements for marine aerosol particles emissions

Sea spray provides the largest mass flux of any aerosol type (**?**) and sea salt aerosol dominates the to-
tal aerosol loading over the remote oceans (**?**). There are few long-term measurement sites of marine
aerosol, all restricted to islands or coastal sites (e.g., MAN, http://aeronet.gsfc.nasa.gov/new_web/maritime_aerosol_network.html).
The source of sea spray aerosol is strongly dependent upon environmental conditions, primarily the

local surface wind speed, but also on wave state (**?**), water temperature, salinity, and the presence of surfactants (**?**). Biological material in the surface water can contribute to a significant organic component to the sea spray aerosol, increasingly so with decreasing particle size (**?**). Most models, however, use simple source functions formulated in terms of wind speed only; the most widely used is that of **?**, which is often applied well beyond the range of conditions from which it was derived and for which it is valid (**?**). **?** found discrepancies between modeled and observed marine aerosol concentrations correlated with sea surface temperature; significant improvement in agreement was found when the model sea spray source function was modified to include a temperature dependence. This result is consistent with a number of laboratory studies which show an increase in coarse mode aerosol production with increasing water temperature (e.g. **????**). Indeed, there appears to be a number of physical and biological effects that can strongly perturb the bubble/aerosol production relationship (**?**).

Extensive in situ measurement of aerosol particles within the marine atmospheric boundary layer is unlikely to be viable. Satellite remote sensing approaches offer the possibility of estimating both ambient aerosol loading and the source flux of marine aerosol. Passive measurement of reflected solar radiation can provide aerosol optical depth (**?**), and some information on both size and vertical distribution (**?**). Active remote sensing can provide much better vertical resolution, and if multiple wavelengths are used, size distributions can be inferred. Both passive and active techniques suffer, however, from the fact that aerosol retrievals are only possible under cloud free conditions. Moreover, complicating matters is that there is more diversity in individual size measurements of sea spray than any other aerosol species (**?**).

The source of sea spray aerosol is breaking waves and the bursting of bubbles generated by them. Many source functions, including that of **?**, scale a production flux of sea spray per unit area white-cap – integrated over its lifetime – by a whitecap fraction parameterized as a function of wind speed. There remains, however, an order of magnitude uncertainty in the parameterization of the whitecap fraction, and there is increasing evidence that neither the production of aerosol per unit area whitecap nor the lifetime of a whitecap are independent of the scale of wave breaking or other water properties (**?; ?; ?, ??**). Recent work on satellite retrievals of the whitecaps (**???**) shows significant promise as a means of providing this driving parameter for sea spray source functions, and implicitly accounting for the wide range of important controlling factors in addition to wind speed (**??**). It might, also, ultimately allow a source function to be specified directly in terms of the satellite measurements. While such an approach would provide near global coverage, the temporal sampling interval is dependent on satellite orbit.

The combination of satellite based estimate of both aerosol loading and source flux offer the optimum means of constraining operational model representation of marine aerosol. Future progress depends on improvements to, and validation of, the retrievals, and on improved estimates of the dependence of sea spray production on wave breaking and water properties. Measurements at very

high wind speeds are also required to better constrain the parameterized source functions under extreme conditions, when sea spray production is greatest, for example during hurricanes or tropical storms.

### 3.4  User requirements for anthropogenic and biogenic aerosols emissions

What is generally perceived as anthropogenic air pollution is in fact a result of complex and poorly understood photochemical processing as well as emissions from point and area sources. Often, anthropogenic emissions are taken to be those associated with domestic, industrial, and mobile sources. However, agricultural emissions, including fertilizers and open maintenance burning, are inconsistently included in the terms biogenic and anthropogenic, respectively. This ambiguity can be initially handled by accepting that, from an aerosol point of view, it is all a single class of processes and "anthropogenic" and "biogenic" emissions follow a similar processing in models. Gridded emissions inventories are commonly generated for primary particles (e.g. primary organic matter (POM), and black carbon (BC)). Sulfates, nitrates, other inorganics, secondary organic aerosol (SOA), and BC are supplemented by emissions of key gases important for secondary aerosol particle production (e.g., $SO_2$, NO$x$, ammonia, isoprene, alkenes, aromatics, terpenes, etc.). These inventories are the result of large scale land classification maps, fuel inventories, and transportation corridor databases. Individual "source" classifications vary by study author, but often include power production, heavy industry/smelting, domestic and biofuel, mobile sources, road dust, agricultural field emissions, agricultural/domestic stack and burn piles as well as plant emissions of such species as isoprene and terpenes. We hold as distinct larger open biomass burning, including agriculture field burning.

Aerosol particle sources are usually prescribed from compiled emission inventories. Despite the efforts put in emission inventories by the community and continuous progress, there remain inherent difficulties in producing accurate inventories. This is because of a number of reasons such as the large variety of point and diffuse sources, uncertainties in emission factors, unknown or unaccounted for sources as well as the model emission approach that is applied (**?**). Among emission uncertainties, there even exist a hierarchy of errors. While point and area sources are less uncertain year after year thanks to satellite data, emission factors remain uncertain due to the impossibility to measure them in "realistic" conditions and due to their strong dependence on the environment. Moreover satellite-based inventories may miss "small" sources as it is the case for smoke inventories in agricultural burning regions.

Since the error on emission inventories automatically translates into a similar or even larger error in concentrations, a user requirement on emission uncertainties might be tempting. However it should be kept in mind that uncertainties and biases in emissions are difficult to estimate and reducing the error to a single number might not be possible. Aerosol source inversion techniques (e.g., **??**) have made some progress but are not yet at a stage where they can constrain emission invento-

ries to better than the user requirement. Such studies can nevertheless point to regional problems in emission inventories.

One ideally requires emission inventories that have a resolution as good as the model resolution. For global modeling systems, this amounts to a spatial resolution and sampling of typically 50 km, although of course many benefits in modeling aerosol transport and deposition may be gained by running NWP at high resolution, even if sources are not known at that resolution. As computing power increases, it is relatively easy to increase model resolution. Sub-grid scale information in emission inventories can be use to post-process and downscale, at least statistically, the simulated model concentrations (**?**). New methods based for example on population density as a proxy are also being used (**?**). For these reasons, it is appropriate that global emission inventories always aim for spatial resolution and sampling that are higher than that of models at a given time (i.e., we recommend a minimum of 10 km resolution given the current state of play). Even higher resolutions (< 1km) are required for regional and urban air quality models given that the typical scale for emissions is very small (e.g., the width of a road for surface traffic).

Temporal distribution of emission inventories can be critical as emission inventories need to sample the diurnal, weekly and seasonal cycles in emissions. Since some aerosol data products are only available at day (e.g., AOD retrieved in the visible part of the electromagnetic spectrum), it is important to deal with the diurnal cycle in emissions so as not to introduce biases in the simulated quantities. As modeling improves, it may become necessary to move from static gridded inventories to include feedbacks with societal (e.g., public holidays, agricultural practices) or meteorological (e.g., influence of cold spells on emissions from heating/biofuel systems or dry spells/wind on stack burning) conditions. Biogenic emissions from plants have also a strong dependency on temperature and water stress.

Aerosol particle speciation in global aerosol models should be reflected in global emission inventories with a minimum of aerosol precursors such as $SO_2$, $NH_3$, $NOx$, and primary aerosol particles such as elemental or BC and POM. Industrial dust and fly ash are often left out but can be important in some regions (as China), and should be included in user requirements. Requirements on speciation for volatile organic compounds (VOCs) are more difficult to set out because it is unclear what level of complexity is required in global aerosol models whose aim is to reproduce mass or number concentrations or optical thickness due to secondary organic aerosols (SOA). We argue here that speciation of VOCs is directly related to the complexity of the aerosol scheme considered and is more difficult to link to user requirements. This is further complicated by SOA production likely being a product of joint anthropogenic emissions. At the minimum bulk seasonal emissions of key classes of reactive VOCs are required (e.g., alkenes, aromatics, isoprene, terpenes).

Aerosol particle properties, such as size and composition, play an important role in determining the aerosol particle radiative efficiency and the ability to serve as cloud condensation nuclei, as well as in having health-related impacts. User requirements on aerosol particle mass or number

size distributions translate into user requirements on aerosol particle size resolution at the emission points. Such user requirements can be expressed in several ways, i.e. on $PM_{10}$, $PM_{2.5}$ and $PM_1$ emission rates, or in combined requirements on aerosol particle mass and number emission rates for typical aerosol size ranges. Historically, the focus has been first on $PM_{10}$, then $PM_{2.5}$ and lastly on $PM_1$ both for health impacts and its connection to cloud formation. The concept itself of PM at a given size cut-off is directly linked to the availability of sampling inlets, but with more current and future instruments we can expect to have a complete information on the aerosol size distribution.

### 3.5 User requirements for open biomass burning aerosols emissions

Biomass burning emissions represent a highly temporally and spatially variable source of aerosols to the atmosphere and reliable and timely estimates are a key input to air quality and atmospheric composition forecasts. Here we define open biomass burning emissions as emissions by fire consuming open vegetation in fields, grasslands or forests. Biofuel or stacked agricultural burning are included in the anthropogenic and biogenic emissions and not considered as open biomass burning. Within the International Global Atmospheric Chemistry (IGAC) project, there is at present about half a dozen advanced global aerosol models that include emissions from vegetation fires that could be included in a multi-model ensemble forecast. Only half of the IGAC-participating models are currently operational, the others remain in a research and development stage (for some, only the vegetation fire component is not operational).

Several real-time smoke forecasting products exist, and are related to satellite based active fire hot spot or burn area databases. The most established global aerosol forecasts are represented in the International Cooperative in Aerosol Prediction (ICAP). Four models include dedicated smoke treatment: CAMS (ECMWF and partners), MASINGAR (MRI-JMA), GEOS-5 (NASA), NAAPS (US Navy), where the first two use emissions from the Global Fire Assimilation System (GFAS, **???**), the second from a similar Fire Radiative Power (FRP) based Quick Fire Emissions Database (QFED, **?**), and the last from the hot spot-based Fire Locating and Modeling of Burning Emissions (FLAMBE) system (**?**). Currently most models scale biomass burning emissions to reach acceptable values of biomass burning aerosol optical thickness close to observations (MODIS or AERONET). This scaling factor ranges from 1.7 for the Met Office Unified Model limited area model configuration over South America that was used for the SAMBBA campaign (**?**) to 1.8-4.5 for GEOS-5 (**?**) and 3.4 for CAMS (**?**). In CAM5 (**?**), regional scaling factors are used (**?**). In small-fire regions, the required factors can be much larger (**?**). The need for these correcting factors arises both from possible underestimation of the biomass burning aerosol emissions and from model biases.

Emissions of aerosols, and other pollutants, associated with open biomass burning are estimated using emission factors which convert between the mass of fuel consumed (derived from FRP or burnt area observations) and the species of interest via the carbon content of the fuel (e.g., **???**). These emission factors are typically calculated using laboratory or field campaign measurements of

smoke constituents which, while providing accurate measurements, may not be fully representative of all biomass burning and smoke conditions. In particular large uncertainties, and missing observations, persist in emission factors for different fuel types (e.g., peat), fire conditions (smoldering vs. flaming), and smoke processing scenarios (e.g., in the presence of clouds, day-time vs. night-time conditions) following, e.g., (**?**). Increased and more extensive in situ measurements of different

fire types would provide the data required to improve emission factors currently used in the operational models. Incorporating meteorological parameters (**??**), such as surface temperature, humidity and soil moisture, which could be done in NRT in the operational models will also be beneficial in adapting otherwise static emission factors to particular environmental conditions. A special case is also provided by peat fires which for their extent and intensity are an important contributor to

global carbon emissions, especially during ENSO related events in Indonesia (for example see the dedicated section in the BAMS State of the Climate 2015 (**?**) or **?**). The remotely detectable signal from peat fires is relatively small and the proportionality to biomass burnt is less certain for these fires than for above-ground fires. Also, the emission factors vary for individual fires so that estimates on a small scale have a limited accuracy. Observations that would help in better constraining the fire

emission factors would be of great usefulness.

There are other ways in which fire emissions can be estimated globally, for example from smoke observations or from burnt areas estimations. These two alternative approaches could not be used in a real time operational framework and have limitations themselves. The uncertainties in emission estimates from smoke observations are still large due to variable and relatively poorly known optical

properties of aerosols, poorly characterized errors of the used atmospheric chemistry and transport models, and noise in the satellite observations. For burnt area products, uncertainties arise mostly from small fires remaining undetected in the burnt area observations and large uncertainties in the estimates of the rather variable input of available fuel load and combustion completeness. For peat fires in particular, the burn depth is not constrained with global observations.

An increase in the number and coverage of observations will certainly improve biomass burning emission estimates. Currently fire products from sensors on low orbit (MODIS, VIIRS) and geostationary satellites (SEVIRI, GOES, Himawari-8) are available. To estimate emissions, observation gaps may occur due to cloud cover or when satellite observations are not available, and the consistent merging of FRP from different satellites is still an open research topic, because their values are often

very different and globally biased. However, combining the high temporal resolution of the geostationary products, which would greatly help in accounting for the usually strong diurnal cycle of fire emissions, and the higher precision and global reach of low Earth orbiting products is an important objective. Future satellite observations might help in reducing the discrepancy between low Earth orbiting and geostationary products.

To support the assessment of fire impacts, measurements of the combustion species (aerosols, reactive and greenhouse gases) are needed. There are several stations that can support verifications

of haze forecast, but their number is very limited and some existing stations do not share data in a timely manner. There is also a network of ground-based observations, including Global Atmosphere Watch (GAW) stations and other global networks (e.g., AERONET). Lidar networks can also help to identify plume heights.

Fire emissions occur most of the time in the Planetary Boundary Layer (PBL). However, for some large fires, estimated at roughly 15% of all fires, fire emissions are released in the free troposphere above the PBL (**???**). In some extreme cases, fire emissions can even reach the Upper Troposphere – Lower Stratosphere region (**?**). The height in the atmosphere at which this occurs is often referred to as the injection height. An observational dataset of injection heights exists through the MISR Plume Height Project (MPHP, **?**), based on a combination of MISR smoke aerosol and MODIS thermal anomaly products. This dataset has recently been updated and extended to produce the MPHP2 dataset. These observations have been very useful in calibrating and/or evaluating global biomass burning emissions injection height datasets (**?**). Satellite products that can provide, in near-real time, an estimate of this injection height would greatly help in accurately forecasting large biomass burning events. Another factor of uncertainty, to a lesser extent, is also the shape of the vertical injection profile. In this case, profiling observations would be required (see also discussion on lidar observations in subsection 3.2).

In this section we have highlighted some of the challenging aspects related to the estimation of emissions from biomass burning. In addition extensive work in drawing user requirements has recently been done by the Interdisciplinary Biomass Burning Initiative (IBBI) and GAW APP-SAG. A draft Concept Note and Expert Recommendations for a Regional Vegetation Fire and Smoke Pollution Warning and Advisory System (RVFSP-WAS) was written, which form the basis of user requirements for biomass burning aerosols (WMO GAW Report No. 235, available at http://www.wmo.int/pages/prog/arep/gaw/documents).

### 3.6 User requirements for removal processes

Wet and dry deposition and sedimentation are important removal processes that control the predication the atmospheric aerosol distribution. However, the aerosol deposition fluxes themselves may become important NAAP forecast products, for example to forecast the soiling of solar panels.

The removal processes are modeled as a function of available meteorological variables describing boundary layer mixing. Wet deposition requires information about the occurrence of convection, precipitation and fog. Dry deposition modeling requires information related to particle size, shape, density and hygroscopicity. It also needs information about the state of the land surface and the vegetation, in particular for soluble aerosols. NAAP takes these meteorological variables from the underlying operational NWP models. It should be noted that improving the forecast of precipitation remains a major challenge for the NWP. Inaccuracies of the precipitation forecast directly influence