# Peer review of "Prediction with a focus on data requirements"

_Atmospheric Chemistry and Physics, 2018_

## Referee Comment (RC1) · Anonymous Referee #2 · 15 Mar 2018

**Review of Benedetti et al**. Status and Future of Numerical Atmospheric Aerosol Prediction

This is a useful review of the status of an important, emerging field, and is sufficiently well developed to be posted as a discussion paper, in my opinion. With some refinement, I think it deserves publication in ACP. The review touches on many things, and it would be helpful if it also provided a bit more synthesis, i.e., if it identified more specifically which issues are currently the most important for certain applications, and which might be achieved most readily.

As this is a review paper, key references might be added in paragraphs where they are lacking, and in several places, text covering a topic in several places could be consolidated, making the presentation more concise and easier to follow. I have included below some specific suggestions for the authors to consider. As I don't have time to reread the paper for this review, I've given it a fairly careful review now. So this is my full review, and as such, it can be posted when the time comes.

1. Section 1.1, Point 1. You might add something along the lines of "reasonably well documented" to the data distribution requirements.
2. Section 1.1, Point 3, line 102. "… for verification and model refinement purposes, …"
3. Section 1.2, lines 118-119. Perhaps: "… data from an observational platform and simulations from a model." Just to distinguish measurements from model results.
4. Section 1.3, line 223. Perhaps: "… aerosol particle emission, secondary production, and removal."
5. Section 2.1. Could you say more about why "ensembles" in general seem to work so much better than individual models? For example, wouldn't the selection of ensemble members matter?
6. Section 3.1, line 278. Might be: "… whereas in remote regions, transports and aerosol processes control the uncertainty."
7. Section 3.1, line 288. By "DA" I assume you mean data assimilation, but this is not defined previously, as best I can tell.
8. Section 3.1, lines 295-296. Might be: "… temporal resolution, speciation, aerosol size distribution, and hygroscopicity." Wouldn't hygroscopicity matter for aerosol-cloud interaction and particle removal efficiency in the models? Perhaps aerosol light-absorption properties is a bit more removed from the considerations for forecasting, but I thought I'd just mention that as a factor that might also be worthy of mention here.
9. Section 3.2, around line 305. I'm a bit confused by this discussion. "Biogenic" aerosol often refers to the aerosol produced by secondary processes from gases emitted by natural vegetation. You seem to mention all these elements in this section, but as written, it is difficult to discern how they relate to each other.
10. Section 3.2.1. Regarding the accuracy of inventories, another issue is "small" sources. For satellite-based inventories, sources too small to be detected by satellite fall into this category. To take one example, this can be a big issue for smoke inventories of agricultural-burning regions.
11. Section 3.2.4. I'm wondering whether the requirements for particle vertical distribution, light-absorbing properties, and/or Mass Extinction Efficiency should be covered in this or one of the other subsections here.
12. Section 3.2.5, lines 377-378. I think there is a typo regarding the subscripts.

13. Section 3.3, Line 400 ff. In small-fire regions, the required factors can be much larger. See, e.g., *Petrenko et al.*, JGR 2017, *doi:10.1002/2017JD026693*.
14. Section 3.3.1, lines 463-464. Overlying smoke opacity and fire emissivity are two additional factors that might be mentioned here (I know they are mentioned elsewhere in this section). A similar point applies to "small" fires (see point 10 above). More generally, the fire-emissions subsections seem a bit longer than necessary – a little reorganization could help consolidate and remove a bit of "rambling." This paper is quite long, which is fine, but it would be a service to readers to consolidate as much as possible.
15. Section 3.3.1. Another, empirical approach relating FRP directly with smoke emission should be mentioned here: *Ichoku, C., and L. Ellison*, 2014. Global top-down smoke-aerosol emissions estimation using satellite fire radiative power measurements. *Atmosph. Chem. Phys., 14, doi:10.5194/acp-14-6643-2014.*
16. Section 3.3.1, Line 501-506. Especially as this is a review paper, some references regarding inverste modeling would be appropriate here.
17. Section 3.3.3, line 564. Another appropriate reference here, for completeness, would be *Val Martin et al.* 2012, Space-based observations constraints for 1-D plume-rise models. *J. Geophys. Res. 117, D22204, doi:10.1029/2012JD018370.*
18. Section 3.4, lines 596-597. It would be helpful here to include one more sentence, saying what they accomplished.
19. Section 3.4, lines 603-604. Would pressure or wind sensors be most efficacious in this case?
20. Section 3.4, lines 629-633. Perhaps this reference would be useful here: *J. Li et al.*, 2016. Reducing Multi-sensor Monthly Mean Aerosol Optical Depth Uncertainty Part I: Objective Assessment of Current AERONET Locations. *J. Geophys. Res.* 122, doi:10.1002/2016JD026308.
21. Section 3.4 overall. I note that the desert dust section goes into much less detail about processes (e.g., surface roughness length, mobilization thresholds, etc.) than the corresponding discussion in the biomass burning section.
22. Section 3.6, line 685. It might be worth mentioning here that dry removal also depends on particle size, shape, density, and hygroscopicity.
23. Section 4.1, first paragraph. To some extent, by assimilating radiances rather than retrieved quantities, all the assumptions and issues treated in the satellite retrievals get pushed onto the model. The assumptions involved are likely to be more consistent, as you note, but not necessarily better, given the attention the modelers must pay to all the other components of the model.
24. Section 4.1, lines 750-757. Are there any appropriate references for this material?
25. Section 4.2, lines 771-777. This largely duplicates the discussion in Section 4.1.
26. Section 4.2.2, lines 797-804. One appropriate reference here might be: *Zhang, J., and J.S. Reid*, 2006. MODIS aerosol product analysis for data assimilation: Assessment of over-ocean level 2 aerosol optical thickness retrievals, *J. Geophys. Res. 111*, doi:10.1029/2005JD006898.
27. Section 4.2.2, Points 1 and 2. AeroCom and AeroSat are spearheading considerable work in the area of pixel-level uncertainties for satellite aerosol retrievals. One example publication: *Witek, M. et al.*, 2018. New approach to the retrieval of AOD and its uncertainty from MISR observations over dark water. *Atmosph. Meas. Tech.* doi.org/10.5194/amt-11-429-2018.

28. Section 4.2.4. In practical terms, is there specific temporal sampling that would address specific data-assimilation needs?
29. Section 4.2.5, lines 842-848. Shouldn't this have been covered, actually in more detail, in the earlier, measurement sections? Also, obtaining pixel-level uncertainties on any retrieval-produced speciation is another issue about which something might be said.
30. Section 4.2.5, lines 865-866. The idea that dust and sea salt are "coarse mode," whereas pollution and smoke are "fine mode," is a gross oversimplification. Both dust and sea salt have size-distribution tails that extend into the fine mode, and often dominate the fine mode. If speciation really matters for the modeling applications under consideration, this needs to be clarified.
31. Section 4.2.5. How good would the particle size and AAOD information need to be to make a significant contribution to NWP?
32. Section 4.2.5, lines 873-874. Surface measurement will not get the transported aerosol, and where there are local aerosol sources, interpreting the results can be complex.
33. Section 4.2.6. MODIS aerosol observations are effectively continued by VIIRS. However, the data records for other instruments, such as CALIPSO and MISR, are at greater risk.
34. Section 4.2.7, lines 901-904. However, errors in a slope derived from two or more spectral AOD measurements can be large. And if there are several modes in the column, interpretation of AE is not straightforward.
35. Section 4. Aside from data assimilation, are there other aerosol forecast applications, and if so, could their requirements be summarized or at least mentioned?
36. Section 5.1, line 951. The residence time of aerosols is "short" compared to… This matters for the discussion here. Transported aerosol can stay aloft for days, even exceeding a week, in the troposphere. Please clarify what is meant here.
37. Section 5.1, lines 957-959. However, spatial and usually temporal sampling by commercial aircraft tends to be highly skewed. You might elaborate on how the limitations affect application to aerosol forecasting.
38. Section 5.1. This is a good summary of network capabilities. It would be helpful for the purposes of the current paper to summarize the strengths and limitations as they relate to aerosol forecasting in particular, e.g., desired site locations, coincident meteorological observations, etc. (Many networks also have a strong climate or air quality focus.)
39. Section 5.2, line 1012. Perhaps, in one place in the paper, you could clarify what is meant by "high temporal resolution" for the applications under consideration, and elsewhere refer to that section for clarification. I get the impression that 3-hourly temporal resolution is desired for most of the applications considered here, but maybe not all.
40. Section 5.3, lines 1044-1048. This seems like a fairly comprehensive list. Are some species higher priority than others, perhaps at different locations? Generally for this section, are some quantities higher priority than others?

---

## Referee Comment (RC2) · Anonymous Referee #1 · 24 Apr 2018

Review "Status and future of Numerical Atmospheric Aerosol Prediction with a focus on data requirements" by Benedetti et al.

This manuscript aims at providing current status of aerosol prediction models, and at reviewing requirements for aerosol observations used by these models. The number of centers providing aerosol forecasting at regional or global scales have been growing significantly over the last decade. It is therefore a useful initiative.

The paper is very interesting and well written. However, there is a strong disparity in quality between sections. The sections at the beginning of the documents are either excellent or very good with only some suggestions listed below. But starting with Section 3.4 things gets too specific. They either focus on the research limited to a specific location, or discuss in details a specific model. Some sections contain incomplete information related to aerosol modeling. As the paper aims at incorporating as much models as possible, I would recommend synthesizing key elements of these models rather than going into the details of strengths and weaknesses of a few models and field campaign. I have also provided below General comments, which may help improve the manuscript.

General comments:
1. It would be useful to introduce mathematically aerosol prediction as an initial and boundary conditions problem as opposed to aerosol projection, which is essentially a boundary condition problem. This will help understand that data assimilation is particularly important for aerosol prediction, while for future projection emissions scenarios are the key factor.
2. At the core of any transport model, there is an advection solver. Models use different solvers, with some creating spurious waves. These numerical oscillations are generally smooth out with a diffusive scheme, creating numerical (unphysical) diffusion. These drawbacks are too important to be ignored, and I would recommend addressing them. An example of discrepancy generated by advection schemes has been discussed by Ginoux (2003). He showed that poor representation of dust size distribution in models was primarily due to the numerical solver of sedimentation.
3. A source of error considered in data assimilation is the inconsistency between simulated and observed variables. This is discussed in the manuscript but what is missing is the description of the dependent variables of the prognostic equations in these models. You should mention that dependent variables of these equations are mass/number concentrations, as it will clarify the discussion, while observations are mostly optical properties. Passing from one to the other necessitates assumptions and consequently error.
4. Is ocean data assimilation not important to be mentioned for seasonal to sub-seasonal aerosol prediction? How could you make any correct aerosol prediction without representing the right phase of large-scale oscillation such as ENSO or NAO? Maybe you should add a sentence about this without developing as it is beyond the scope of the paper.

5. Emission of several aerosols depends strongly on vegetation. For example, biomass burning will obviously depend on the amount of biomass, dust emission is drastically reduced in presence of any vegetation cover, and the emission of biogenic organic precursors depends on vegetation cover. It may be valuable in this paper to include data requirements for vegetation cover, as new model developments often increase the level of interactions between vegetation and aerosols. Evans et al. (2016) showed that dust variability in Australia is amplified by dynamic vegetation in agreement with satellite observations. Also, are there any recommendations to validate land model results used for aerosol prediction?

6. An additional application of aerosol forecasting model is to provide support during field campaign. The model provides direct information on aerosol optical thickness and concentrations for effective flight planning, while feedbacks from measurements constantly evaluate the model for successful model improvements (Chin et al., 2003).

SPECIFIC COMMENTS:

Section 1.2. I would suggest adding some sentences related to above comments 1 to 3.

Section 1.3. You may want to mention the use of forecasting model to support field campaigns (see above comment 6).

Section 3.2.2. Last paragraph. Increasing resolution does not necessarily mean better model skills. It may request new tuning of parameters of subscale processes (e.g. orographic gravity wave drag), as well as larger ensemble runs due to large variability. I wish I could propose a reference related to aerosol, but Kapnick et al. (2018) discusses such issues for the prediction of snow over the western US.

Section 3.4. This section on dust and the following on sea-salt are much shorter than the previous section related to biomass burning. Is there a justification for it? Section 3.4. No discussion on dust sources, which is the base of any dust study and modeling. All dust models use a form or another of preferential dust sources defined by Prospero et al. (2002) and adapted for global models (Ginoux et al., 2001; Tegen et al., 2002; Zender et al., 2003; Ginoux et al., 2012). These source functions were necessary because soil properties from global inventories (e.g. FAO) were and still are unrealistic in arid and hyper-arid regions. Although, model representing the physical processes of dust emission have been around for a long time (e.g. Marticorena and Bergametti, 1995; Shao, 2001), they have to be adapted to accommodate major discrepancy in soil texture datasets, the driver of dust emission. There is the interesting work of objectively compare different dust source inventories (Cakmur et al., 2006). It may be adequate to perform similar exercise with more recent inventories.

Section 3.4. Not one word on soil texture, soil moisture, vegetation cover, and mineralogy, although these are key elements to simulate dust emission, distribution and optical properties. I would recommend including them in a paragraph with references. On the other hand, there is a discussion on the difficulty to represent sub-scale dry and wet convection. These are important processes for dust emission, but it may be better to discuss boundary layer parametrization in a "meteorological" section. Why are you mentioning 3 field campaigns? And these ones in particular, are the others less important?

Section 3.4. Satellite data. You mention IASI but there are more than 2 groups working on retrieving dust from the data. Geostationary satellites have their own quality for aerosol prediction, and SEVIRI has been quite useful to retrieve dust sources (Schepanski et al., 2007), or detect haboobs (Ashpole and Washington, 2013). Also, I would mention the promising results from GRASP algorithm (Chen et al., 2018).

Section 3.5. There is no mention of the temperature dependency of sea salt emission. Most models are now considering it, specifically the parameterization of Jaegle et al. (2011)

Section 3.6. This section is detailing removal processes of one model (NAAP), but they are generally treated quite differently in other models. It reads as a technical report of the NAAP model. Also, it seems that important processes are missing, such as in cloud scavenging, Bergeron process, etc. It would be more useful to learn about the method to parameterize the different physical processes rather than learning what is useful or not to run NAAP.

Section 3.6. Line 703-705. I would mention the work by Yu et al. (2017), which allows evaluating dust deposition by combining MODIS and CALIOP data.

Section 4.2.2 Line 790. Reference(s) would be useful.

Section 4.2.5. This section is again focusing on model (CAMS) to discuss its problems. Why should I care about this model if I am not using it?

Line 874: There is method to derive aerosol speciation from AERONET (Schuster et al., 2005), and more recently there are promising possibilities with GRASP algorithm (Torres et al., 2017)

REFERENCES

Ashpole, I. and Washington, R., 2013. A new high‑resolution central and western Saharan summertime dust source map from automated satellite dust plume tracking. *Journal of Geophysical Research: Atmospheres*, *118*(13), pp.6981-6995.

Cakmur, R. V., R. L. Miller, J. Perlwitz, I. V. Geogdzhayev, P. Ginoux, D. Koch, K. E. Kohfeld, I. Tegen, and C. S. Zender (2006). Constraining the magnitude of the global dust cycle by minimizing the difference between a model and observations." *J. Geophys. Res.,* 111.

Chen, C., Dubovik, O., Henze, D. K., Lapyonak, T., Chin, M., Ducos, F., Litvinov, P., Huang, X., and Li, L.: Retrieval of Desert Dust and Carbonaceous Aerosol Emissions over Africa from POLDER/PARASOL Products Generated by GRASP Algorithm, Atmos. Chem. Phys. Discuss., https://doi.org/10.5194/acp-2018-35, in review, 2018.

Chin, M., P. Ginoux, R. Lucchesi, B. Huebert, R. Weber, T. Anderson, S. Masonis, B. Blomquist, A. Bandy, and D. Thornton (2003), A global aerosol model forecast for the ACE-Asia field experiment, J. Geophys. Res., 108(D23), 8654, doi:10.1029/2003JD003642.

Evans, S., Ginoux, P., Malyshev, S., & Shevliakova, E. (2016). Climate‑vegetation interaction and amplification of Australian dust variability. *Geophysical Research Letters*, *43*(22).

Ginoux, P., Chin, M., Tegen, I., Prospero, J. M., Holben, B., Dubovik, O., & Lin, S. J. (2001). Sources and distributions of dust aerosols simulated with the GOCART model. *J. Geophys. Res.*, *106*(D17), 20255-20273.

Ginoux, P. (2003). Effects of nonsphericity on mineral dust modeling, *J. Geophys. Res.-Atmos., 108*, 4052, doi:10.1029/2002JD002516.

Ginoux, P., Prospero, J.M., Gill, T.E., Hsu, N.C. and Zhao, M., 2012. Global‑scale attribution of anthropogenic and natural dust sources and their emission rates based on MODIS Deep Blue aerosol products. *Reviews of Geophysics*, *50*(3).

Jaeglé, L., Quinn, P. K., Bates, T. S., Alexander, B., and Lin, J. T. (2011). Global distribution of sea salt aerosols: new constraints from in situ and remote sensing observations. *Atmospheric Chemistry and Physics*, *11*(7), 3137-3157.

Kapnick, S. B., Yang, X., Vecchi, G. A., Delworth, T. L., Gudgel, R., Malyshev, S., ... & Margulis, S. A.: Potential for western US seasonal snowpack prediction, *Proceedings of the National Academy of Sciences*, 201716760, 2018.

J. M. Prospero, P. Ginoux, O. Torres, S. E. Nicholson, and T. E. Gill (2002). Environmental characterization of global sou7rces of atmospheric soil dust identified with the nimbus 7 total ozone mapping spectrometer (TOMS) absorbing aerosol product, Rev. Geophys., 40(1), 1002, doi:10.1029/2000RG000095.

Schepanski, K., Tegen, I., Laurent, B., Heinold, B., & Macke, A. (2007). A new Saharan dust source activation frequency map derived from MSG‑SEVIRI IR‑channels. *Geophy. Res. Lett.*, *34*(18).

Schuster, G.L., Dubovik, O., Holben, B.N. and Clothiaux, E.E., 2005. Inferring black carbon content and specific absorption from Aerosol Robotic Network (AERONET) aerosol retrievals. *Journal of Geophysical Research: Atmospheres*, *110*(D10).

Torres, B., Dubovik, O., Fuertes, D., Schuster, G., Cachorro, V.E., Lapyonok, T., Goloub, P., Blarel, L., Barreto, A., Mallet, M. and Toledano, C. (2017). Advanced characterisation of aerosol size properties from measurements of spectral optical depth using the GRASP algorithm. *Atmospheric Measurement Techniques*, *10*(10), p.3743.

Zender, C. S., Bian, H., and Newman, D. (2003). Mineral Dust Entrainment and Deposition (DEAD) model: Description and 1990s dust climatology. *Journal of Geophysical Research: Atmospheres*, *108*(D14).

Shao, Y. (2001). A model for mineral dust emission. *Journal of Geophysical Research: Atmospheres*, *106*(D17), 20239-20254.

---

## Author Comment (AC1) · 20 Jun 2018

First of all, we would like to thank the reviewer for his/her in-depth review of the paper and the useful comments. The paper has been extensively re-written aiming at giving the full picture with the utmost clarity. Please see below for detailed answers to the reviewer's suggestions.

*1. Section 1.1, Point 1. You might add something along the lines of "reasonably well documented" to the data distribution requirements.*
*2. Section 1.1, Point 3, line 102. "... for verification and model refinement purposes, ..."*
*3. Section 1.2, lines 118-119. Perhaps: "... data from an observational platform and simulations from a model." Just to distinguish measurements from model results.*
*4. Section 1.3, line 223. Perhaps: "... aerosol particle emission, secondary production, and removal."*

The suggested sentences have been added/corrected.

*5. Section 2.1. Could you say more about why "ensembles" in general seem to work so much better than individual models? For example, wouldn't the selection of ensemble members matter?*

For aerosols, Sessions et al (2015) and more recently Xian et al (2018, submitted to QJRMS) have shown that the ensemble is the top performer. Selection of members does not matter as long as individual members are truly independent and of roughly equal skill.

*6. Section 3.1, line 278. Might be: "... whereas in remote regions, transports and aerosol processes control the uncertainty."*

Added.

*7. Section 3.1, line 288. By "DA" I assume you mean data assimilation, but this is not defined previously, as best I can tell.*

This has been defined.

*8. Section 3.1, lines 295-296. Might be: "... temporal resolution, speciation, aerosol size distribution, and hygroscopicity." Wouldn't hygroscopicity matter for aerosol-cloud interaction and particle removal efficiency in the models? Perhaps aerosol light-absorption properties is a bit more removed from the considerations for forecasting, but I thought I'd just mention that as a factor that might also be worthy of mention here.*

Hygroscopicity is indeed very important as it determines the aerosol optical properties and affects assimilation of, for example, AOD and in turn, the aerosol forecasts. It is now mentioned explicitly in the text.

*9. Section 3.2, around line 305. I'm a bit confused by this discussion. "Biogenic" aerosol often refers to the aerosol produced by secondary processes from gases emitted by natural vegetation. You seem to mention all these elements in this section, but as written, it is difficult to discern how they relate to each other.*

We tried to reflect the fact that there is no agreed nomenclature that encompasses all "anthropogenic" aerosols. "Biogenic" can also refer to natural occurring primary, secondary or to aerosol emitted from anthropogenic sources which have certain organic materials in them. We agree that it is confusing. From the point of view of the paper, we just treat this category as distinct from "natural" aerosols such as sea salt and dust. **Please note that now this is section 3.4**.

**10. Section 3.2.1. Regarding the accuracy of inventories, another issue is "small" sources. For satellite-based inventories, sources too small to be detected by satellite fall into this category. To take one example, this can be a big issue for smoke inventories of agricultural-burning regions.**

The following sentence has been added to acknowledge the problem: "Moreover satellite-based inventories may miss "small" sources as it is the case for smoke inventories in agricultural burning regions."

**11. Section 3.2.4. I'm wondering whether the requirements for particle vertical distribution, light-absorbing properties, and/or Mass Extinction Efficiency should be covered in this or one of the other subsections here.**

We agree with the reviewer that section 3.2.4 or other subsections of 3.2 could have been used to cover particle vertical distribution, light absorbing properties, and/or Mass Extinction Efficiency. However, we consider it to be more appropriate to cover these in the following sections since section 3.2 describes gridded emission inventories which estimate emission by combining activity data with emission factors, and thus the mentioned parameters are not relevant to estimate the emissions.

The afore mentioned parameters are addressed in the manuscript as follows:

Vertical distribution: in lines 497-508 of section 3.3.3 to highlight the importance of injection height in Fire emissions and then 509 and 510 to mention to importance of the injection profile and the need for observations constraining this source of uncertainty. Lines 845-851, in section 4.2.3, also stress the need for information of the vertical particle structure through vertically resolved observations. Finally, lines 1099-1103, in section 5.2, emphasise on the need to combine vertically integrated observations with vertically resolved and surface observations.

Extinction: lines 870-872 in section 4.2.5 to underline the importance of the extinction of single species in order to estimate their radiative impact.

Absorption properties: lines 884-886 of section 4.2.5 to highlight its use to constrain absorbing aerosols in the model. Lines 1129 to 1133 of section 5.3 reveals its need (together with other aerosol optical properties) to evaluate the direct and semi-direct effect on aerosol absorption properties.

**12. Section 3.2.5, lines 377-378. I think there is a typo regarding the subscripts.**

Typo corrected.

**13. Section 3.3, Line 400 ff. In small-fire regions, the required factors can be much larger. See, e.g., Petrenko et al., JGR 2017, doi:10.1002/2017JD026693.**

Reference to Petrenko et al. (2017) has been added. The sentence now reads:
"In small-fire regions, the required factors can be much larger (Petrenko et al, 2017).
**Please note that this is now section 3.5**.

*14. Section 3.3.1, lines 463-464. Overlying smoke opacity and fire emissivity are two additional factors that might be mentioned here (I know they are mentioned elsewhere in this section). A similar point applies to "small" fires (see point 10 above). More generally, the fire-emissions subsections seem a bit longer than necessary – a little reorganization could help consolidate and remove a bit of "rambling." This paper is quite long, which is fine, but it would be a service to readers to consolidate as much as possible.*

This comment has been addressed with a radical consolidation of the section on biomass burning (now section 3.5).

*15. Section 3.3.1. Another, empirical approach relating FRP directly with smoke emission should be mentioned here: Ichoku, C., and L. Ellison, 2014. Global top-down smoke-aerosol emissions estimation using satellite fire radiative power measurements. Atmosph. Chem. Phys., 14, doi:10.5194/acp-14-6643-2014.*

The reference has been added, thank you.

*16. Section 3.3.1, Line 501-506. Especially as this is a review paper, some references regarding inverse modeling would be appropriate here.*

Reference to Huneeus et al 2012 and Escribano et al 2017 have been added.

*17. Section 3.3.3, line 564. Another appropriate reference here, for completeness, would be Val Martin et al. 2012, Space-based observations constraints for 1-D plume-rise models. J. Geophys. Res. 117, D22204, doi:10.1029/2012JD018370.*

The reference has been added, thanks.

*18. Section 3.4, lines 596-597. It would be helpful here to include one more sentence, saying what they accomplished.* [i.e. "A first step in creating such a network was undertaken during the recent Fennec project (Hobby et al., 2013; Roberts et al., 2017)."]

**Please note that now this is section 3.2.**
We have added text to address this point:

"A first step in creating such a network was undertaken during the recent Fennec project, which deployed stations in 2011 (Hobby et al 2013): the deployed stations could not be maintained beyond 2013 (Roberts et al 2017), so do not provide continuous monitoring or a long climatology, but have demonstrated that (i) reporting the sub 3-minute variance in winds is generally unimportant, but resolving the diurnal cycle is critical, (ii) there are substantial biases even in analysed winds, which miss the summer-time wind maximum in the central Sahara, and (iii) that it is important to evaluate

dust uplift together with model winds, and that observational records of this relationship are invaluable (Roberts et al, 2018). "

**19. Section 3.4, lines 603-604. Would pressure or wind sensors be most efficacious in this case?** [ i.e. A much denser network of high-quality pressure observations is needed to better constrain models in this regard]

We have added text to address this question,
"A much denser network of high-quality pressure and wind observations is needed to better constrain models in this regard. Pressure measurements have the advantage of being less affected by local (and often sub-grid) conditions (e.g. topographic circulations, inhomogeneities in roughness) than wind measurements, and have, through data assimilation, a far greater impact on the analysed heat low which, in turn, controls the model winds.

**20. Section 3.4, lines 629-633. Perhaps this reference would be useful here: J. Li et al., 2016. Reducing Multi-sensor Monthly Mean Aerosol Optical Depth Uncertainty Part I: Objective Assessment of Current AERONET Locations. J. Geophys. Res. 122, doi:10.1002/2016JD026308.**

Thanks. We have added this reference.

**21. Section 3.4 overall. I note that the desert dust section goes into much less detail about processes (e.g., surface roughness length, mobilization thresholds, etc.) than the corresponding discussion in the biomass burning section.**

We have added significant new text to address these points, covering vegetation and roughness and dust sources.

**22. Section 3.6, line 685. It might be worth mentioning here that dry removal also depends on particle size, shape, density, and hygroscopicity.**

Hygroscopicity has been added in the list of variables.

**23. Section 4.1, first paragraph. To some extent, by assimilating radiances rather than retrieved quantities, all the assumptions and issues treated in the satellite retrievals get pushed onto the model. The assumptions involved are likely to be more consistent, as you note, but not necessarily better, given the attention the modelers must pay to all the other components of the model.**

We fully agree with the reviewer on this point. This is how the sentence reads:

"The optimality of assimilating retrieved aerosol products versus radiances and the choice of a suitable algorithm or method for fast radiative transfer in the shortwave are still being debated. On the one hand direct radiance assimilation avoids the problem in the diversity between the model and the retrieval assumptions (aerosol type, refractive index, meteorological parameters, etc,), on the other hand the complexity of the observations might complicate or even prevent the implementation of radiance assimilation, especially for advanced sensors such as multi-angle instruments or polarimeters. In the end, the most pragmatic approach prevails in an operational

context, hence the assimilation currently depends heavily on the availability of good quality retrieval products with reliable uncertainty estimates."

**24. Section 4.1, lines 750-757. Are there any appropriate references for this material?**

Several references on the various assimilation approaches have been added.

**25. Section 4.2, lines 771-777. This largely duplicates the discussion in Section 4.1.**

This has been reworded to avoid duplication.

**26. Section 4.2.2, lines 797-804. One appropriate reference here might be: Zhang, J., and J.S. Reid, 2006. MODIS aerosol product analysis for data assimilation: Assessment of over-ocean level 2 aerosol optical thickness retrievals, J. Geophys. Res. 111, doi:10.1029/2005JD006898.**

The reference has been added.

**27. Section 4.2.2, Points 1 and 2. AeroCom and AeroSat are spearheading considerable work in the area of pixel-level uncertainties for satellite aerosol retrievals. One example publication: Witek, M. et al., 2018. New approach to the retrieval of AOD and its uncertainty from MISR observations over dark water. Atmosph. Meas. Tech. doi.org/10.5194/amt-11-429-2018.**

The reference has been added, thanks for the suggestion.

**28. Section 4.2.4. In practical terms, is there specific temporal sampling that would address specific data-assimilation needs?**

This is a difficult question to answer. We tried to be constructive without being too prescriptive. This is how the section now reads:

"The issue of temporal resolution is similar to that of spatial resolution. In principle high-temporally resolved data are beneficial to the analysis, particularly because they provide information on the diurnal aerosol variability. However, issues connected to large data volume may arise. This is particularly true for datasets coming from geostationary satellites which are now providing data at with temporal resolution of 10-15 minutes. In some cases, such data have to be heavily thinned or averaged (Saide et al 2014). This is obviously only a technical limitation which might not be applicable across the range of assimilation systems. For example, the new generation of Japanese geostationary satellites, Himawari 8-9 (Bessho et al 2016) provides excellent data that have been demonstrated to be of use for data assimilation (Yumimoto et al 2016). For ground-based instruments, similar considerations can be made, although data volume might not be as high."

**29. Section 4.2.5, lines 842-848. Shouldn't this have been covered, actually in more detail, in the earlier, measurement sections? Also, obtaining pixel-level uncertainties on any retrieval-produced speciation is another issue about which something might be said.**

All sections have been heavily rewritten for clarity and readability. This has been addressed in the most recent version of the manuscript.

**30. Section 4.2.5, lines 865-866. The idea that dust and sea salt are "coarse mode," whereas pollution and smoke are "fine mode," is a gross oversimplification. Both dust and sea salt have size-distribution tails that extend into the fine mode, and often dominate the fine mode. If speciation really matters for the modeling applications under consideration, this needs to be clarified.**

The oversimplification is a by-product of modelling a complex natural phenomena at the global scale with the use of parameterizations. Of course, nature is a continuum and the distinction between coarse and fine particles is purely academic.

**31. Section 4.2.5. How good would the particle size and AAOD information need to be to make a significant contribution to NWP?**

From the paper: "The accuracy of AAOD would need to be comparable to that of total AOD for the product to have an impact in the analysis." Said this, any inclusion of AAOD would be beneficial.

**32. Section 4.2.5, lines 873-874. Surface measurement will not get the transported aerosol, and where there are local aerosol sources, interpreting the results can be complex.**

We mention in the paper that the ground-based network should be relatively dense. Of course, this will still be limited to over-land. However, we feel that the surface measurements have an important role to play.

**33. Section 4.2.6. MODIS aerosol observations are effectively continued by VIIRS. However, the data records for other instruments, such as CALIPSO and MISR, are at greater risk.**

This is unfortunate. We comment on this and refer to follow-on mission lidar such as Aeolus and EarthCARE that can help alleviate the problem.

**34. Section 4.2.7, lines 901-904. However, errors in a slope derived from two or more spectral AOD measurements can be large. And if there are several modes in the column, interpretation of AE is not straightforward.**

This has been acknowledged. The paper now reads: "On the other hand, errors in a slope derived from two or more spectral AOD measurements can be large. Moreover interpretation of AE is not straightforward in a column where several aerosol modes are present. The usefulness of AE over AOD (or fine/coarse mode AOD) is still a matter of debate in the retrieval and assimilation communities."

**35. Section 4. Aside from data assimilation, are there other aerosol forecast applications, and if so, could their requirements be summarized or at least mentioned?**

The focus of the paper is on user requirements for operational aerosol prediction, and data assimilation is one of the tools to improve this prediction. We do not consider data assimilation as an end in its own but rather as a mean.

**36. Section 5.1, line 951. The residence time of aerosols is "short" compared to... This matters for the discussion here. Transported aerosol can stay aloft for days, even exceeding a week, in the troposphere. Please clarify what is meant here.**

The sentence has been expanded:
"Since the atmospheric residence time of aerosol particles in the troposphere is relatively short (from hours to ~1 week, depending on species-specific physical processes and meteorological conditions) and the footprint area of a single station may be limited, there is a need for ground-based observation networks with sufficient density and representativeness of stations."

**37. Section 5.1, lines 957-959. However, spatial and usually temporal sampling by commercial aircraft tends to be highly skewed. You might elaborate on how the limitations affect application to aerosol forecasting.**

This is acknowledged in the text: "Data collected from commercial aircraft can provide invaluable observations for model evaluation (e.g., In-service Aircraft for a Global Observing System, IAGOS; http://www.iagos.org/). At the moment, however, this is not established for operational aerosol applications. Moreover due to the spatial and temporal skewness of the distribution of data collected from aircraft (often more dense close to airports), some care needs to be put into assimilating them into operational systems."

**38. Section 5.1. This is a good summary of network capabilities. It would be helpful for the purposes of the current paper to summarize the strengths and limitations as they relate to aerosol forecasting in particular, e.g., desired site locations, coincident meteorological observations, etc. (Many networks also have a strong climate or air quality focus.)**

The following sentences have been added to the section 5.1:
"Referring to the requirements of observations as outlined in the introduction (i.e., ease of access/ consistency, uncertainty, and speed of delivery), globally consistent and available datasets such as for AOD from AERONET or NASA satellite by default currently drive the evaluation process and consequently model development. AERONET's ability to provide high accuracy of fine and coarse mode AOD data over the globe with typical preliminary data availability within 6-24 hours makes it a favored metric variable (Sessions et al., 2015). Likewise, the maturity, coverage, speed and ease of access of MODIS aerosol retrievals makes MODIS AOD retrievals the dominate satellite verification product (as discussed in section 4 favored for data assimilation as well). This dominance of AOD to some degree is to the exclusion of perhaps more applicable baseline variables not meeting the noted observational requirement, such as PM2:5 /PM10 or aerosol vertical distribution. As discussed in section 3, additional evaluation variables related to model microphysics (chemical composition, absorption, size, full solar and IR radiative properties, etc.) are only sporadically available, and rarely collected simultaneously."

And in this section is also mention that "it is mandatory to provide additional information on the observation site with a correct classification based on its spatial representation (regional or global) and its localization (environment types and emission types)."

Additionally in section 5.3 it is mentioned:

"To fully understand processes, more sites with co-located observations of visibility, cloud, radiation, vertical profiles of temperature, relative humidity as well as winds and aerosol properties would be highly desirable. Precipitation and deposition observations are also extremely relevant for benchmarking. Innovative designs for global measurement systems (existing technological platforms such as commercial aircraft, cell phones, cars, etc.) should be further exploited."

*39. Section 5.2, line 1012. Perhaps, in one place in the paper, you could clarify what is meant by "high temporal resolution" for the applications under consideration, and elsewhere refer to that section for clarification. I get the impression that 3-hourly temporal resolution is desired for most of the applications considered here, but maybe not all.*

This is considered in the most recent version of the manuscript.

*40. Section 5.3, lines 1044-1048. This seems like a fairly comprehensive list. Are some species higher priority than others, perhaps at different locations? Generally for this section, are some quantities higher priority than others?*

It is clear that aerosol dominant species depends on the region but considering the focus of the manuscript is global aerosol prediction models all the aerosol chemical species must be in the list. We can rephrase the sentence but when you have chemical samples, you can consider all the species. By priorities we would like to say that all the observations are important but would include first size distribution, then, number concentration and finally, chemical speciation.

---

## Author Comment (AC2) · 20 Jun 2018

First of all we would like to thank the reviewer for his/her in-depth review of the paper and the useful comments. All additional reference suggested by the review were added to the manuscript. The paper has been extensively re-written aiming at giving the full picture with the utmost clarity. Please see below for detailed answers to the reviewer's suggestions.

***General comments:***

***1. It would be useful to introduce mathematically aerosol prediction as an initial and boundary conditions problem as opposed to aerosol projection, which is essentially a boundary condition problem. This will help understand that data assimilation is particularly important for aerosol prediction, while for future projection emissions scenarios are the key factor.***

This has been done in the section "Aerosol Prediction Models".

"Moreover, although some of the data requirements presented here are shared with aerosol models for climate applications, here we focus on numerical aerosol prediction at the short and medium-range (up to 10 days). In this context, we are essentially dealing with an initial and boundary condition problem for which the requirements for assimilation have high priority. For sub-seasonal to seasonal aerosol prediction, which is not dealt with here specifically, requirements on ocean state and variability are also important as well requirements for the development of prognostic emission models. In the wider context of aerosol projections for climate prediction, the emphasis is much more on emission scenarios and the requirements will consequently be different."

***2. At the core of any transport model, there is an advection solver. Models use different solvers, with some creating spurious waves. These numerical oscillations are generally smooth out with a diffusive scheme, creating numerical (unphysical) diffusion. These drawbacks are too important to be ignored, and I would recommend addressing them. An example of discrepancy generated by advection schemes has been discussed by Ginoux (2003). He showed that poor representation of dust size distribution in models was primarily due to the numerical solver of sedimentation.***

In the section "Aerosol Prediction Models" we added the following sentence:

"Each [model] relies on different dynamical cores, advection solvers, and aerosol microphysics schemes that necessarily generate a large degree of diversity among the various models (see for example Ginoux, (2003)). The range of horizontal and vertical resolutions across the models is also very diverse, as is inline versus offline architecture."

***3. A source of error considered in data assimilation is the inconsistency between simulated and observed variables. This is discussed in the manuscript but what is missing is the description of the dependent variables of the prognostic equations in these models. You should mention that dependent variables of these equations are mass/number concentrations, as it will clarify the discussion, while observations are mostly optical properties. Passing from one to the other necessitates assumptions and consequently error.***

This sentence has been added in the section "Data assimilation for aerosol prediction".

"As discussed previously, most aerosol assimilation systems rely at the moment on products such as AOD, rather than raw measurements such as satellite radiances. However, the tendency in the future will likely be towards the use of satellite radiances, either raw or aggregated and possibly cloud-cleared, for consistency with the current approach in NWP. This represents a challenge for both the model developers and the data providers and might also involve joint development of observation operators. The last point is particularly true considering that there is a fundamental

inconsistency between simulated and observed variables. The prognostic variables in the model are the mass/number concentrations of the individual species whereas the observed variables are mostly optical properties. Converting from one to the other necessitates assumptions and consequently is a source of error which has to be mitigated."

**4. Is ocean data assimilation not important to be mentioned for seasonal to sub-seasonal aerosol prediction? How could you make any correct aerosol prediction without representing the right phase of large-scale oscillation such as ENSO or NAO? Maybe you should add a sentence about this without developing as it is beyond the scope of the paper.**

This has been included in the reply to comment 1, and reported above. We should stress that at this point most aerosol prediction systems are focussing on the short to medium-range, with most systems predicting up to day 5. Sub-seasonal and seasonal aerosol prediction is very much at its beginning. Most of the requirements will be similar to the medium-range prediction, but of course the complexity of the system will increase and the role of other drives such as ocean initialization will become more important.

**5. Emission of several aerosols depends strongly on vegetation. For example, biomass burning will obviously depend on the amount of biomass, dust emission is drastically reduced in presence of any vegetation cover, and the emission of biogenic organic precursors depends on vegetation cover. It may be valuable in this paper to include data requirements for vegetation cover, as new model developments often increase the level of interactions between vegetation and aerosols. Evans et al. (2016) showed that dust variability in Australia is amplified by dynamic vegetation in agreement with satellite observations. Also, are there any recommendations to validate land model results used for aerosol prediction?**

While this is a very valid point, we feel that to add requirements on vegetation cover would go beyond the scope of the paper. In the case of the dust forecast, one option is to include NRT data (with one day lag) from MODIS vegetation index in the source map instead a vegetation climatology, that it is most commonly used. Another thing is to consider this vegetation changes in the meteorological solver.

**6. An additional application of aerosol forecasting model is to provide support during field campaign. The model provides direct information on aerosol optical thickness and concentrations for effective flight planning, while feedbacks from measurements constantly evaluate the model for successful model improvements (Chin et al., 2003).**

This has been added to the section "Aerosol Prediction Models".

"These systems are used for various applications, including, but not exclusively, global air quality forecasts (dust and biomass burning), operation impacts, boundary conditions for regional systems and flight campaign planning (Chin et al., 2003).

**SPECIFIC COMMENTS:**

**Section 1.2. I would suggest adding some sentences related to above comments 1 to 3.**

This has been addressed.

**Section 1.3. You may want to mention the use of forecasting model to support field campaigns (see above comment 6).**

See above.

*Section 3.2.2. Last paragraph. Increasing resolution does not necessarily mean better model skills. It may request new tuning of parameters of subscale processes (e.g. orographic gravity wave drag), as well as larger ensemble runs due to large variability. I wish I could propose a reference related to aerosol, but Kapnick et al. (2018) discusses such issues for the prediction of snow over the western US.*

We agree with the reviewer. This sentence has been added in the section "Aerosol Prediction Models":

"In general, increasing resolution does not necessarily mean better model skills as it may request new tuning of parameters of subscale processes (e.g. orographic gravity wave drag), as well as larger ensemble runs due to high variability."

We did not include the reference as not directly relevant.

*Section 3.4. This section on dust and the following on sea-salt are much shorter than the previous section related to biomass burning. Is there a justification for it? Section 3.4. No discussion on dust sources, which is the base of any dust study and modeling. All dust models use a form or another of preferential dust sources defined by Prospero et al. (2002) and adapted for global models (Ginoux et al., 2001; Tegen et al., 2002; Zender et al., 2003; Ginoux et al., 2012). These source functions were necessary because soil properties from global inventories (e.g. FAO) were and still are unrealistic in arid and hyper-arid regions. Although, model representing the physical processes of dust emission have been around for a long time (e.g. Marticorena and Bergametti, 1995; Shao, 2001), they have to be adapted to accommodate major discrepancy in soil texture datasets, the driver of dust emission. There is the interesting work of objectively compare different dust source inventories (Cakmur et al., 2006). It may be adequate to perform similar exercise with more recent inventories.*

**This section is now 3.2**
We agree that these aspects were under-emphasised in the submitted paper and have added paragraphs that address these points. Moreover the other sections were also shortened and re-organized to make the whole paper more homogeneous and readable.

"For a reliable prediction of mineral dust aerosol, sufficiently accurate knowledge of both the emitting soil and the deflating winds is needed. Both aspects suffer from insufficient observational constraints, creating a large challenge for quantitative emission predictions. Important source regions globally include the Sahara/Sahel, Southwest Asia/Middle East, Taklimakan/Gobi deserts of China, Australia and the Southwest United States/adjacent Mexico (Prospero et al 2002). However, larger source regions show substantial fine structure and throughout the world there are also many individual sources such as in Patagonia, the Arctic plains, and countless dry or drying lake beds. Estimating dust emission sources can also be performed from satellite data (for examples see Huneeus et al (2012), Schutgens et al (2012), Yumimoto and Takemura (2013), Escribano et al (2016), Escribano et al (2017), Di Tomaso et al (2017)).

Dust models typically employ maps of dust source functions (e.g. Zender et al (2003), Ginoux et al (2012)), because soil properties in arid and hyper-arid regions from global inventories are insufficient to provide consistent soil texture information. This includes aspects such as soil particle size distribution and binding energies but also the existence of roughness elements and soil moisture content that impact on mobilization thresholds. See Darmenova (2009) for a comprehensive review. This severely limits the level of complexity that can be put into models representing the physical processes of dust emission (e.g. Marticorena and Bergametti (1995), Shao et al (2001), Kok et al

(2014)). In order to get a better understanding of the involved uncertainties, an update to the objective comparison of different dust source inventories by Kakmur et al (2006) would be desirable and could be extended to take into account uncertainties in the dust emission parameterization itself."

***Section 3.4. Not one word on soil texture, soil moisture, vegetation cover, and mineralogy, although these are key elements to simulate dust emission, distribution and optical properties. I would recommend including them in a paragraph with references.***

We have added substantial text on this,
"In addition to that, dust emission is further complicated by suppressing influences of soil moisture (Fecan et al, 1998) and vegetation cover, including brown vegetation from a previous rainy period (Kergoat et al, 2017), which can vary on relatively small time and spatial scales. This is particularly acute in the semi-arid Sahel with its seasonal vegetation, also creating large variations in surface roughness (Cowie et al, 2013). There is currently a debate to what extent the mineralogy of emitted dust particles should be taken into account, as this would alter both its interactions with radiation (Journet etal 2014) and cloud microphysics. While certainly an interesting field of research, the former aspect is probably more relevant on longer timescales, while the latter is not even considered in most current dust prediction models."
"Finally, the dust-focused satellite data should be complemented by improved space-born assessments of soil moisture and vegetation cover (green and brown) to better characterize varying conditions in source regions (Kergoat et al 2017)."

***On the other hand, there is a discussion on the difficulty to represent sub-scale dry and wet convection. These are important processes for dust emission, but it may be better to discuss boundary layer parametrization in a "meteorological" section. Why are you mentioning 3 field campaigns? And these ones in particular, are the others less important?***

"The wind requirements for dust modelling are quite distinct from those of other components of the model, since the uplift occurs as a result of very rare high wind-speed events, the "tail" of the very wind distribution. The importance of processes such as the day-time breakdown of the LLJ and haboobs are specific to dust modelling as such we have found that discussion of these issues is best placed in the dust section, rather than a general meteorological section.
In the original draft we refer to the AMMA, Fennec, BoDeX and JADE field campaigns (as well as the CV-DUST project and Cape Verde Observatory) as these have deployed networks of stations, in some cases long-lived, in remote areas similar to those that we believe are required to improve dust in NWP. The authors have been involved in other less campaigns, such as GERBILS and SAMUM, which we have not cited as they are less relevant here, but would welcome other specific suggestions for campaigns we have missed.

***Section 3.4. Satellite data. You mention IASI but there are more than 2 groups working on retrieving dust from the data. Geostationary satellites have their own quality for aerosol prediction, and SEVIRI has been quite useful to retrieve dust sources (Schepanski et al., 2007), or detect haboobs (Ashpole and Washington, 2013). Also, I would mention the promising results from GRASP algorithm (Chen et al., 2018).***

We have added a reference to Klüser et al., 2012 for IASI.
We already discuss use of geostationary infrared data, but have added a note on its use for source detection, "Infrared products are being developed but still have biases related to atmospheric

moisture (Banks et al., 2013). These would need to be further improved and provided in near-real time for data assimilation, but have been useful for source detection (Schepanski et al., 2007)."
NWP needs information on the land surface, dust and dust-generating winds, independent of the uplift mechanism, so we have not discussed haboob detection.
We have added a reference to GRASP,
"or those produced with the GRASP algorithm (Chen et al., 2018) are promising but have more limited space-time coverage."

**Section 3.5. There is no mention of the temperature dependency of sea salt emission. Most models are now considering it, specifically the parameterization of Jaegle et al. (2011)**

This sentence has been added to the section on sea salt emissions (**now section 3.3**).

"Jaegle et al (2011) found discrepancies between modelled and observed marine aerosol concentrations correlated with sea surface temperature; significant improvement in agreement was found when the model sea spray source function was modified to include a temperature dependence. This result is consistent with a number of laboratory studies…"

**Section 3.6. This section is detailing removal processes of one model (NAAP), but they are generally treated quite differently in other models. It reads as a technical report of the NAAP model. Also, it seems that important processes are missing, such as in cloud scavenging, Bergeron process, etc. It would be more useful to learn about the method to parameterize the different physical processes rather than learning what is useful or not to run NAAP.**

NAAP is not a specific model, but stands for Numerical Atmospheric Aerosol Prediction.

**Section 3.6. Line 703-705. I would mention the work by Yu et al. (2017), which allows evaluating dust deposition by combining MODIS and CALIOP data.**

The following sentence has been added:

"Recently, Yu et al (2015) tried to infer dust deposition by combining MODIS and Cloud-Aerosol Lidar with Orthogonal Polarization (CALIOP) data. Their CALIOP-based multi-year mean estimate of dust deposition matches better with estimates from in situ measurements and model simulations than previous satellite-based estimates."

**Section 4.2.2 Line 790. Reference(s) would be useful.**

References have been added.

**Section 4.2.5. This section is again focusing on model (CAMS) to discuss its problems. Why should I care about this model if I am not using it?**

Specific mentions to the CAMS model have been removed.

**Line 874: There is method to derive aerosol speciation from AERONET (Schuster et al.,2005), and more recently there are promising possibilities with GRASP algorithm (Torres et al., 2017)**

References have been added. The sentence now reads:

"Wherever direct speciation measurements are possible, those would be best suited to be used to correct model prediction of a given aerosol species. These could be measurements derived from a (relatively dense) network of ground-based instruments and/or from satellites. Some promising

results to derive aerosol speciation from AERONET observations have been obtained by Schuster et al. (2005) and more recently by Torres et al. (2017) using the Generalized Retrieval of Aerosol and Surface Properties (GRASP) algorithm.